# Risk assessment of debris flow disaster based on the cloud model—Probability fusion method

**Li Li** [1]*, **Bo Ni**[1], **Yue Qiang**[1], **Shixin Zhang**[2], **Dongsheng Zhao**[1], **Ling Zhou**[1]

**1** Department of Civil Engineering, Chongqing Three Gorges University, Wanzhou, China, **2** Department of Earth Sciences, Chengdu University of Technology, Chengdu, China

* lily6636694@163.com

**Data Availability Statement:** All relevant data are within the paper and its Supporting Information files.

## Abstract

This paper proposes a new debris flow risk assessment method based on the Monte Carlo Simulation and an Improved Cloud Model. The new method tests the consistency of coupling weights according to the characteristics of the Cloud Model firstly, so as to determine the weight boundary of each evaluation index. Considering the uncertain characteristics of weights, the Monte Carlo Simulation is used to converge the weights in a minimal fuzzy interval, then the final weight value of each evaluation index is obtained. Finally, a hierarchical comprehensive cloud is established by the Improving Cloud Model, which is used to input the comprehensive expectation composed of weights to obtain the risk level of debris flow. Through statistical analysis, this paper selects Debris flow scale ($X_1$), Basin area ($X_2$), Drainage density ($X_3$), Basin relative relief ($X_4$), Main channel length ($X_5$), Maximum rainfall ($X_6$) as evaluation indexes. A total of 20 debris flow gullies were selected as study cases (8 debris flow gullies as model test, 12 debris flow gullies in reservoir area as example study). The comparison of the final evaluation results with those of other methods shows that the method proposed in this paper is a more reliable evaluation method for debris flow prevention and control.

## 1 Introduction

Debris flow is a sudden geological disaster that seriously threatens the life and property safety of residents which is a special flood that occurs between valleys in mountainous areas. It is known as one of the main geophysical killers in mountainous areas [1]. Therefore, the risk assessment of debris flow is of great practical significance for project site selection, project prevention and control, disaster emergency plan formulation [2].

The basic research on debris flow risk assessment was first carried out by Varnes in the United States [3]. Liu et al. proposed a classic formula for the classification of debris flow risk levels in single ditch [4] by determining the primary and secondary indices. For a long time, people have carried out systematic research on the risk assessment of debris flow. For example, traditional AHP [5], Extension Theory [6], Grey Theory [7], etc. With the current computer computing performance improvement, some new algorithms are applied in this field. Cao et al. applied 3S technology to collect data and field survey data, and constructed a cloud

**Funding:** This work was supported by the Scientific and Technological Research Program of Chongqing Municipal Education Commission (Grant No. KJQN202001218, KJQN202101206, KJQN202201238), Research development and application of "big data intelligent prediction and early warning cloud service platform for geological disasters in the Three Gorges Reservoir Area" of Chongqing Municipal Education Commission (HZ2021012),the Open fund of Chongqing Three Gorges Reservoir Bank Slope and Engineering Structure Disaster Prevention and Control Engineering Technology Research Center (SXAPGC21ZD01), the Science and technology innovation project of Chongqing Wanzhou District Bureau of science and technology (wzstc20210305), Nanjing 2022 "Science and Technology Three Gorges" Chongqing Wanzhou District counterpart support project of Chongqing Wanzhou District Bureau of science and technology(2022101S-02), Special key program of Chongqing Technology Innovation and Application Development (Grant No. cstc2019jscx-tjsbX0015), and "Chongqing Huanjiang structure disaster prevention and reduction theory and key technology" of Chongqing University Innovation Research Group (201928). The funders had no role in study design, data collection and analysis, decision to publish, or preparation of the manuscript.

**Competing interests:** The authors have declared that no competing interests exist.

model for debris flow risk assessment [8]. Banihabib et al. proposed a Bayesian network model (BNs) to point out effective predictors for identifying the risk of debris flow events [9]. Li et al. used the rock engineering system (RES) to determine weights and applied the rock engineering system combined with the fuzzy C-means method (RES-FCM) in the mud risk assessment [10]. Gu et al. introduced the Projection Pursuit Classification (PPC) of acceleration genetic algorithm to evaluate the risk level of debris flow hazards [11]. With the development of geographic information technology, some new methods based on the high-precision monitoring system have been realized. For example, Niu et al. applied ArcGIS spatial analysis function to classify landslide hazards by superimposing the information layers calculated by different methods [12]; Hou and Du used the Geographic Information System (GIS) and the Remote Sensing (RS) techniques to design a ponding dispersion algorithm, and then convert the volume of water accumulation in the catchment area into the depth of it and submerged range [13].

For risk assessment, except a few intelligent algorithms, weight calculation is the most important problems. Common weight calculation methods include the Maximum Dispersion Method [14], the Analytic Hierarchy Process [15], the Entropy Method [16], the Projection Pursuit Method [17], the Technique for Order Preference by Similarity to Ideal Solution [18] and so on. These methods can be divided into subjective and objective categories. In particular, objective methods usually have large deviations due to the difference of information entropy and calculation process. Therefore, obtaining reasonable weights has always been a key concern in the field of risk assessment [19,20]. In recent years, some new weight fusion methods [21,22] have been proposed, and some scholars have done a lot of research on the multi-source information fusion method of classical Dempster-Shafer evidence theory [23,24] to improve the reliability of weight calculation. In summary, multi-source information fusion has been a hot research topic in the field of evaluation. What is more, the debris flow risk assessment in this paper is a typical multi-criteria problem.

However, in previous studies, the error of weight values selected by different methods is often lack of consistency test. Therefore, this paper refers to the method of testing consistency with the cloud model characteristics proposed by Li et al [25] to calculate the weight boundary of each evaluation index. As a classical random number simulation method, the Monte Carlo Simulation has been extended to the fields of medicine [26], material science [27], chemistry [28] and so on. In order to adapt to the fuzzy attribute of multi-criteria problems, this paper uses the Monte Carlo Simulation Method to select the weight value within the tested weight boundary, so as to obtain a reasonable and fuzzy final weight value. Considering the extensibility and reliability of the Cloud Model in the fuzzy decision-making field [29–31], we construct a normal comprehensive cloud model to expand the coverage at the grade boundary, thus obtaining the final risk classification.

Based on the Improved Cloud Model and the Monte Carlo Simulation Method, this paper fully explores the cloud model characteristics to test of coupling weights and complete the risk classification of debris flow. Section 2 of this paper gives an overview of the study area. Section 3 introduces the selection of evaluation indexes and the construction process of the model. Section 4 analyzes the calculation results. And section 5 discusses the rationality and advantages of the proposed method in detail.

## 2 The overview of study area

In this paper, 8 typical debris flow gullies proposed by Kuang et al. were selected for model test [6], and 12 debris flow gullies near Wudongde Hydropower Station [21] were selected for example study. Given that the 8 test samples are widely distributed in location and have been

studied repeatedly, making them reliable data that have stood the test of time, detailed topographic feature will not be described here.

## (1) Wudongde Hydropower Station

The Wudongde Hydropower Station is located in the lower reaches of the Jinsha River in the first step of a four-step hydropower stations. It began to store water after 8 years of construction from 2012 to 2020, and the water level is controlled at 965 metres during initial operation The topography of this region is complicated, large altitude difference between upstream and downstream. Furthermore, there are abundant hydraulic resources. Following the impoundment of the reservoir, the fluctuation of upstream and downstream water level poses a threat to the stability of both the banks and surrounding slopes.

## (2) Geographical location and landform of the study area

The study area is located in the Jinsha River Basin in the upper reaches of the Yangtze River in China, where contains the Hengduan Mountains with separate seasons, the Yunnan-Guizhou Plateau where temperature are as stable as in spring and the eastern Tibet Plateau with perennial low temperature. The special geographical conditions have caused obvious climate differences, making the region form complex landforms such as alpine shrubs, mid-mountain forests, plateau deserts, and meadows.

## (3) Physical geography and terrain characteristics of the study area

The study area of this paper is situated in the Jinsha River basin at the junction of Sichuan and Yunnan provinces with an altitude difference between the upper and lower reaches of more than 3300 meters. It receives its water from precipitation, upper-reaches ice and snow melt water, and groundwater, making it rich in hydraulic resources. The 12 example gullies are distributed in the Yunnan-Guizhou Plateau altitude of 1000–2000 meters, where geological fracture and uplifting tectonic activities are significant. Due to the fast flow velocity and strong erosion, there are abundant sources on both sides of the valley, and the slope on both sides are mostly more than 40 degrees, with the steepest being close to 90 degrees. Furthermore, there are extensive accumulation fans in the tributary estuary, some of which are loose and unstable, collapse and landslide are common.

# 3 Materials and methods

## 3.1 The index system of debris flow disaster

Due to the differences in debris flow scale, research scope and location, debris flow risk assessment can be regarded as a complex and interdisciplinary problem [32]. Therefore, the selected evaluation indexes should have representative and clear physical meaning, and each evaluation index should be independent of each other and easy to quantify. According to the characteristics of debris flows in Southwest China, referring to Huo's statistical results of 42 domestic and foreign literatures [33], the top 6 evaluation indexes with the highest frequency were selected as the evaluation indexes of this paper. They are the debris flow scale ($X_1$), the basin area ($X_2$), the drainage density ($X_3$), the basin relative relief ($X_4$), the main channel length ($X_5$), the maximum rainfall ($X_6$).

The specific meanings of the above six debris flow evaluation indexes are as follows:

$X_1(10^4 m^3)$: The debris flow scale is one of the direct factors to reflect the risk of debris flow. The larger the scale is, the larger the volume of debris flow is, the greater the harm caused by debris flow disaster is, and the greater the potential possibility of debris flow disaster is.

$X_2$(km$^2$): The basin area represents the catchment area surrounded by a watershed. The basin area is positively correlated with the regional sediment yield, which affects the loose solid reserves within a certain range, thus indirectly affecting the volume of debris flow and the potential possibility of debris flow disasters.

$X_3$(km/km$^2$): The drainage density is one of the indirect factors reflecting the hazard of debris flow, which represents the ratio of the total length of gullies to the watershed area. This index indirectly reflects the degree of weathering and sediment yield of rocks in the basin. In general, the larger the cutting density of the basin, the larger the volume of solid loose matter and the larger the liquid runoff, and the greater the potential destructive power when debris flow disasters occur.

$X_4$(km): The basin relative relief represents the height difference between the lowest point and the highest point. The larger the height difference indirectly reflects the poor stability of the mountain, the greater the potential energy of the debris flow disaster, and the faster the flow speed of the loose objects. It is one of the important factors reflecting the destructive ability of debris flow.

$X_5$(km): The main channel length refers to the plane projection distance from the starting point of the main gully to the gully mouth in the basin, which also indirectly reflects the size of the basin area. Within a certain range, the longer the length is, the more loose objects such as sediment are collected along the way, and the greater the destructive power is when debris flow disasters occur.

$X_6$(mm): The maximum daily rainfall is usually considered to be the direct factor triggering debris flow disasters.

Model test and study area example of a total of 20 debris flow gully data are shown in Table 1.

**Table 1. Debris flow ditch data.**

| Samples | Evaluation indexes | | | | | |
|---|---|---|---|---|---|---|
| | $X_1$(×10$^4$m$^3$) | $X_2$(km$^2$) | $X_3$(km/km$^2$) | $X_4$(km) | $X_5$(km) | $X_6$(mm) |
| A | 195.1 | 47.1 | 23.8 | 2.19 | 12 | 102 |
| B | 105 | 53.1 | 21.2 | 2.92 | 18.35 | 97 |
| C | 7.8 | 10.61 | 12.8 | 1.66 | 4.61 | 100.4 |
| D | 10 | 14.1 | 17.8 | 1.94 | 8 | 100.4 |
| E | 1.8 | 4.2 | 12.3 | 1.19 | 2.1 | 100.4 |
| F | 82 | 18.05 | 22.8 | 1.66 | 11.8 | 100.4 |
| G | 9 | 28.32 | 15.6 | 2.8 | 9.05 | 100.4 |
| H | 5 | 3.28 | 22 | 1.13 | 2.3 | 98 |
| Zhugongdi | 4 | 6.5 | 6.24 | 1.34 | 4.98 | 112.5 |
| Yindi gully | 18.1 | 60.5 | 5.08 | 2.25 | 20.16 | 131.2 |
| Jianshanbao | 6.09 | 16.78 | 3.6 | 1.06 | 6.02 | 161.2 |
| Xiushui creek | 3.5 | 8.58 | 6.9 | 1.67 | 2.2 | 181.1 |
| Shuilvqing | 19.3 | 37.1 | 6.72 | 1.74 | 10.51 | 114.5 |
| Shaoshui gully | 4.4 | 3.6 | 9.4 | 1.34 | 3.92 | 161.5 |
| Kuashanqing | 5.74 | 1.18 | 15.5 | 1.29 | 2.78 | 118.5 |
| Yanshui creek | 3.33 | 4.87 | 9 | 1.91 | 4.36 | 112.5 |
| Longtang gully | 5.36 | 2.18 | 13.7 | 1.03 | 2.54 | 151.5 |
| Lufang plain | 6.45 | 1.23 | 19.3 | 1.04 | 2.11 | 115.4 |
| Hujia gully | 14.1 | 8.62 | 6.34 | 1.53 | 5.16 | 118.2 |
| Laohu plain | 20.68 | 5.41 | 44.3 | 1.48 | 4.21 | 111.1 |

Note: Numbers A~H are the 8 debris flow gullies for model test.

**Table 2. Evaluation index level.**

| Risk level | Evaluation indexes | | | | | |
|---|---|---|---|---|---|---|
| | $X_1 (\times 10^4 m^3)$ | $X_2 (km^2)$ | $X_3 (km/km^2)$ | $X_4 (km)$ | $X_5 (km)$ | $X_6 (mm)$ |
| I | $\leq 1$ | $\leq 0.5$ | $\leq 5$ | $\leq 0.2$ | $\leq 1$ | $\leq 25$ |
| II | 1~10 | 0.5~10 | 5~10 | 0.2~0.5 | 1~5 | 25~50 |
| III | 10~100 | 10~35 | 10~20 | 0.5~1.0 | 5~10 | 50~100 |
| IV | $\geq 100$ | $\geq 35$ | $\geq 20$ | $\geq 1.0$ | $\geq 10$ | $\geq 100$ |

Referring to the classic formula for the risk classification of single-ditch debris flow proposed by Liu et al. [4], the 6 evaluation indexes are divided into 4 grades: low risk (I), medium risk (II), high risk (III), extremely high risk (IV). As shown in Table 2.

## 3.2 Model construction

The overall flow chart of Section 3.2 is shown in Fig 1:

The method proposed in this paper uses the cloud model's characteristics to test the consistency of coupling weight intervals, and then it calculates the final weight value using the Monte Carlo Simulation. On this basis, the Cloud Model Theory is used again to construct a comprehensive cloud model to complete the debris flow risk classification.

**3.2.1 Definition and digital characteristic of cloud.** Due to the characteristics of randomness and ambiguity, cloud model has been used in some uncertain problems after its derivation and improvement [8,34]. The cloud model controls the shape of the model through three digital features (Expectation *Ex*, Entropy *En*, and Hyper-entropy *He*), and generates a

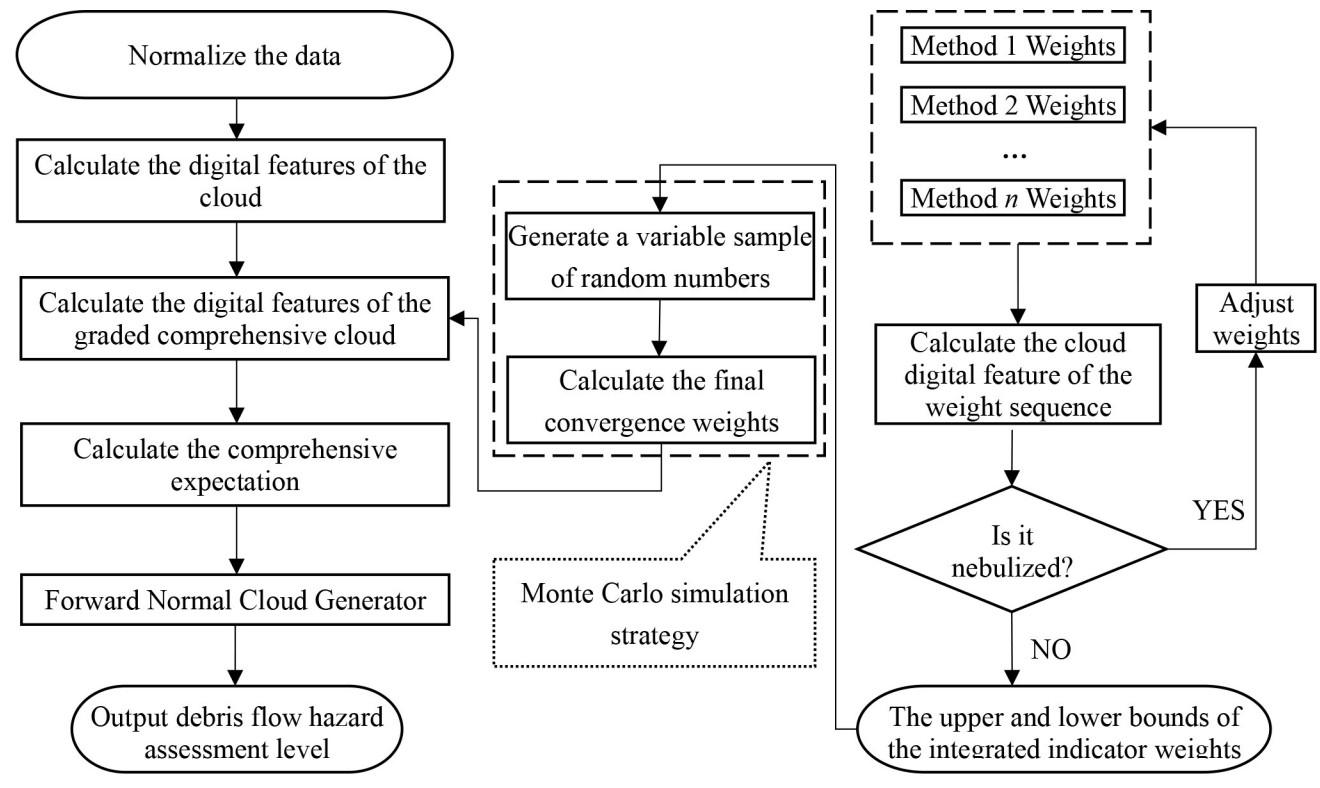

**Fig 1. Model flowchart.**

large number of cloud droplets to form a cloud droplet group through a cloud generator, thereby, the uncertainty conversion between qualitative concepts and quantitative representations can be realized. The cloud is defined as follows:

Let $Z$ be a quantitative interval of a qualitative concept, if $x \in Z$, $x$ represents a random realization of a qualitative concept in interval $Z$, each x in the interval $Z$ corresponds to a membership degree $u(x) \in [0,1]$ with a stable tendency [25]. Then all cloud droplets distributed on $Z$ are collectively called cloud.

The three digital features of the cloud control the interval and shape of the cloud, where Expectation $E_x$ reflects the center of gravity of cloud droplets; Entropy $E_n$ reflects the degree of dispersion and ambiguity of conceptual extension; Hyper-entropy $H_e$ represents the re-description of ambiguity. As shown in Fig 2:

The three digital characteristic of the cloud control its the range and shape, where Expectation $E_x$ reflects the barycenter of the cloud droplets cluster. The entropy $E_n$ reflects the degree of dispersion and ambiguity of the concept extension. In addition, the Hyper-entropy $H_e$ represents a re-description of fuzziness.

**3.2.2 Forward normal cloud generator.** If the point $x$ belonging to the $Z$ interval is substitutied into the cloud generator, the distribution of the membership degree $u(x)$ belonging to a qualitative concept can be generated. This is the Forward Cloud Generator used in this paper, which generates cloud droplets based on Expectation $E_x$, Entropy $E_n$ and Hyper-entropy $H_e$, thereby achieving qualitative to quantitative mapping.

The operation steps of Forward Cloud Generator [35] are as follows:

1. Calculate the Expectation $E_x$, Entropy $E_n$ and Hyper-entropy $H_e$ and determine the number $N$ of cloud droplets according to the relevant data and calculation method;

2. Generate a normal random number $y_i$ with $E_n$ as the expectation and $H_e$ as the standard deviation;

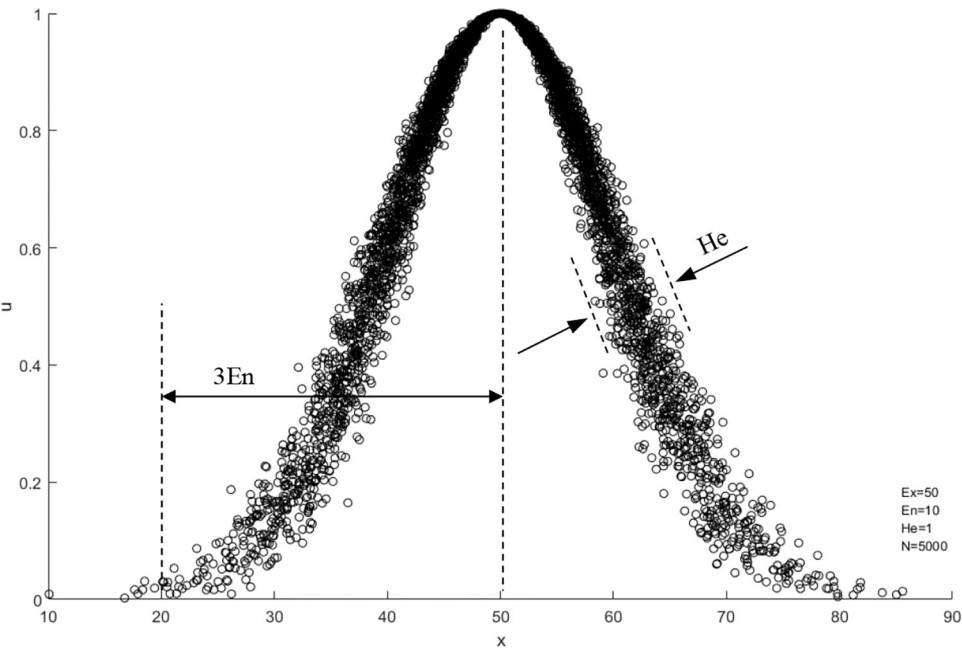

**Fig 2. Cloud digital features.**

3. Generate a normal random number $x_i$ with $E_x$ as the expectation and $y_i$ as the standard deviation;

4. Calculate the membership degree belonging to a qualitative concept:

$$\begin{cases} u(x_i) = 1 - 0.001x \in [E_{x\max}, +\infty) \\ u(x_i) = \exp\left(-\dfrac{(x_i - Ex)^2}{2y_i^2}\right)x \in \text{other interval} \end{cases} \tag{1}$$

5. Obtain the cloud droplet group composed of $N$ cloud droplets and the membership degree $u(x_i)$.

**3.2.3 Weight fusion test method based on cloud atomization.** How to integrate the weights of different methods is the key to obtain reasonable prediction results. The weights calculated by different methods may vary greatly in value due to different emphases, and the results obtained by direct coupling calculation tend to have corresponding deviations. Most of the current weight fusion methods lack the process of checking the reasonableness of the obtained weights. Based on the basic theory of cloud model and cloud digital characteristics, this paper conducts consistency test on the selected weights, and then fuses the weights that meet the conditions after the test. The specific process is as follows:

Refer to the relevant literature to obtain the weights calculated by $W_1$, $W_2$,. . .,$W_n$ in a total of $n$ different methods, and bring into the following Eq [36] to calculate the cloud digital features of each factor weight sequence:

$$\begin{aligned} E_x &= \frac{1}{n}\sum_{i=1}^{n}w_i \\ E_n &= \sqrt{\frac{\pi}{2}}\frac{1}{n}\sum_{i=1}^{n}|w_i - E_x| \\ H_e &= \sqrt{\frac{1}{n-1}\sum_{i=1}^{n}(w_i - E_x)^2 - E_n^2} \end{aligned} \tag{2}$$

In the formula, the meaning of each parameter is the same as that in Section 2.2.1.

After calculating the above cloud digital features, input the forward cloud generator to generate the normal cloud droplet group. As mentioned above, this process is mainly to test the rationality and consistency of the selected weights. By judging the degree of atomization of the cloud droplet group, that is, the greater the degree of atomization, the greater the discrete degree of the selected weight sequence, and the greater the conflict of the weight sequence. Therefore, the final comprehensive weight is obtained by eliminating the atomization phenomenon of the cloud droplet group. Referring to the extensive statistical analysis of the atomization process by Liu et al. [37], it is determined that the cloud starts to atomize when $H_e > E_n/3$, that is, the critical condition is $H_e = E_n/3$. as shown in Fig 3.

It can be seen from Fig 3 that when the ratio of $H_e$ to $E_n/3$ gradually becomes smaller, the cloud droplets cluster gradually begins to converge and take shape.

According to the 6 evaluation indicators selected in Section 2.1, this paper refers to a large number of literature [5–7,21,38–41] and obtains the weights calculated by 8 different methods (Table 3).

Each time, select $n$ ($n \geq 3$) weight values calculated by different methods from Table 3, and bring them into Eq (2) to calculate the cloud digital features $E_n$ and $H_e$ until the critical value

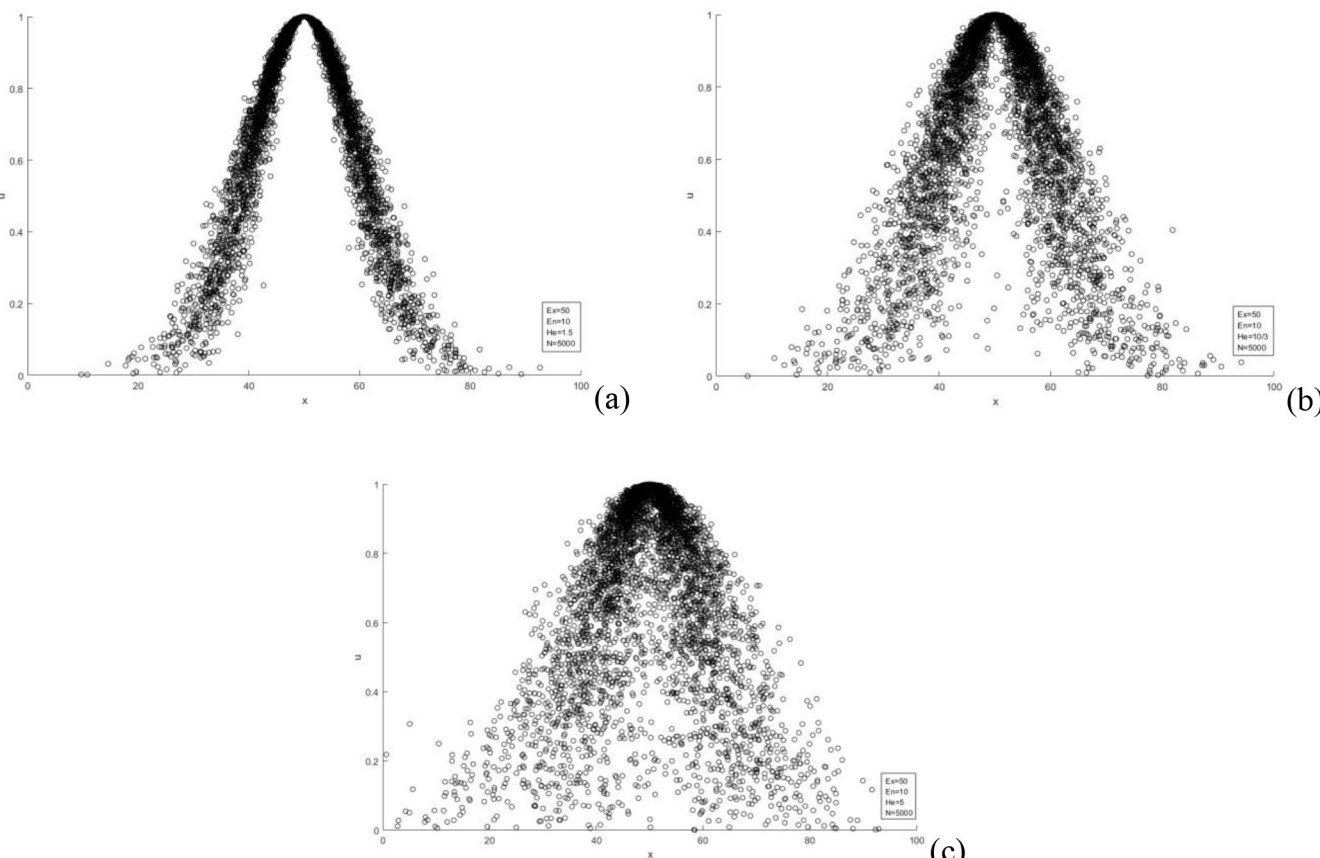

**Fig 3.** Cloud model fogging process: (a) Ex = 50, He = 1.5; (b) Ex = 50, He = 10/3; (c) Ex = 50, He = 5.

of $H_e < E_n/3$ is satisfied, the cloud fogging phenomenon is eliminated. At this time, the $n$ weight values obtained after passing the test can be considered to be consistent. According to the above process, this paper finally selects the weight values calculated by three different methods that satisfy the consistency test from Table 3 (Fig 4).

It can be seen from Fig 4 that the weight change trend after the consistency test is similar, and the weight error of each index is within a small interval.

**Table 3. Evaluation index weight.**

| Evaluation indexes | | | | | | Reference sources |
|---|---|---|---|---|---|---|
| $X_1(\times 10^4 m^3)$ | $X_2(km^2)$ | $X_3(km/km^2)$ | $X_4(km)$ | $X_5(km)$ | $X_6(mm)$ | |
| 0.3458 | 0.2326 | 0.1814 | 0.0790 | 0.1087 | 0.0525 | Ref. Liu et al. 2012 [5] |
| 0.2977 | 0.1649 | 0.1160 | 0.1328 | 0.1298 | 0.1588 | Ref. Luo et al. 2008 [7] |
| 0.3878 | 0.0816 | 0.0612 | 0.1429 | 0.1224 | 0.2041 | Ref. Qiang et al. 2017 [21] |
| 0.3571 | 0.2302 | 0.1789 | 0.0753 | 0.1103 | 0.0481 | Ref. Liu et al. 2010 [38] |
| 0.3643 | 0.1972 | 0.0805 | 0.0805 | 0.0805 | 0.1971 | Ref. Wang et al. 2014 [39] |
| 0.3321 | 0.1749 | 0.1250 | 0.1308 | 0.1590 | 0.0783 | Ref. Wang et al. 2016 [34] |
| 0.3557 | 0.1778 | 0.1555 | 0.1111 | 0.1333 | 0.0667 | Ref. Kuang et al. 2006 [6] |
| 0.2759 | 0.1724 | 0.1379 | 0.1207 | 0.1034 | 0.1897 | Ref. Ren et al. 2013 [41] |

Note: For the literature with more evaluation indexes, 6 evaluation indexes used in this paper are selected for normalization.

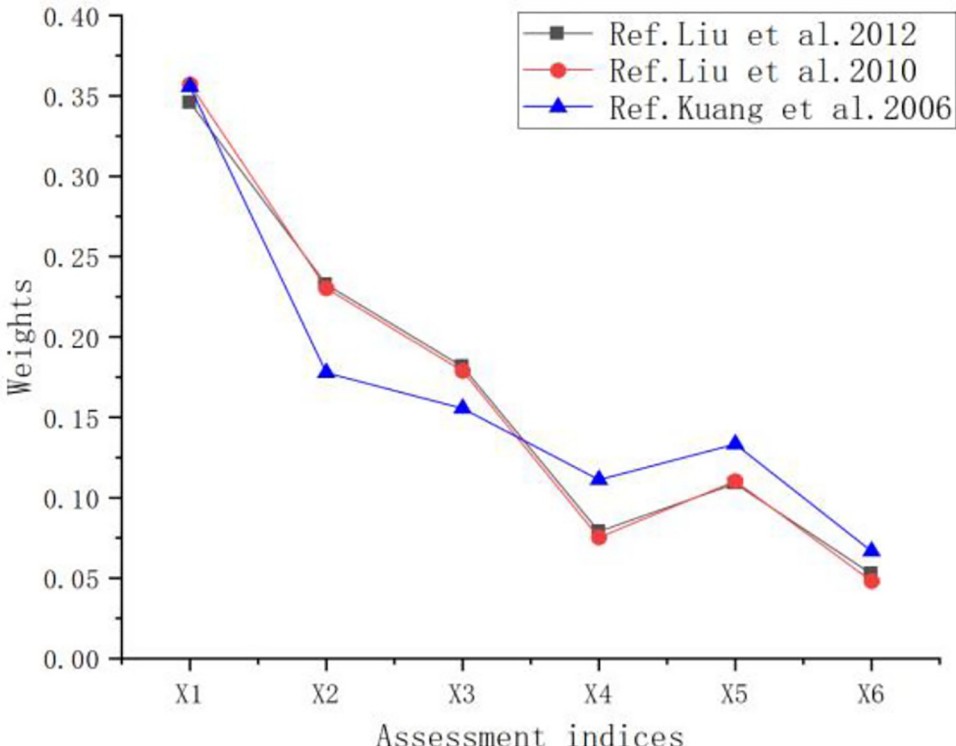

**Fig 4. Weight value satisfying consistency.**

According to the 6 evaluation indexes selected in Section 2.1 and the 3 weight values that satisfy the consistency conditions selected in the above steps, the weight value range of each evaluation index is obtained:

$$w_1 \in [0.3458, 0.3571],$$
$$w_2 \in [0.1778, 0.2327],$$
$$w_3 \in [0.1555, 0.1814],$$
$$w_4 \in [0.0753, 0.1111],$$
$$w_5 \in [0.1087, 0.1333],$$
$$w_6 \in [0.0481, 0.0667].$$

(3)

**3.2.4 The monte carlo stochastic simulation.** Monte Carlo is a stochastic simulation method. According to the law of large numbers, when the sample size is large enough, the occurrence frequency of an event is its probability [42]. As computer performance improves, this theory is often used to solve complex random problems [43]. For example, integrals for solving complex univariate functions. As follows:

Suppose a univariate function formula $f(x)$ is given to calculate the integral $I = \int_a^b f(x)dx$.

1. Randomly select $l$ numbers from the interval [a, b] and name them $x_1, \cdots, x_l$.

2. Calculate $Q_l = (b - a) \cdot \frac{1}{l} \sum_{i=1}^{l} f(x_i)$.

3. The calculated $Q_n$ can be approximately regarded as the integral result of $I = \int_a^b f(x)dx$.

This process can be regarded as solving the area of irregular figure, and when we give up multiplying by the integral interval *(b-a)*, $Q_l$ degenerates into a value in the very small interval within the integral interval. As the number of samples *l* increases, it gradually approaches the midpoint of the interval. After many experiments and verification, we found that after 200 random samplings, $Q_l$ basically stably converges to a very small interval. Combined with the weight boundaries of each evaluation index in Eq (3) in Section 2.2.3, the final weight calculation formula is as follows:

$$f(x_i) = w_{n_{min}} + (w_{n_{max}} - w_{n_{min}}) * rand(l, 1) \tag{4}$$

$$w_n^* = \frac{1}{\Delta l^*} \sum_{min l^*}^{max l^*} \frac{1}{\Delta l} \sum_{i=1}^{l} f(x_i) l^* \in (190, 200] \tag{5}$$

In the formula, *i* is the number of random sampling ($i \in 1,2,\cdots,l$); *n* is the number of evaluation indexes ($n \in 1,2,\cdots,6$); the *rand* function means to return a random real number between [0,1].

According to the above calculation process, as shown in Eq (5), this paper selects the last 10 groups $w_n^*$ after the convergence of each evaluation index to calculate the average value, and obtains the final weight $w^* = (0.352605\ 0.205313\ 0.169655\ 0.09345\ 0.121884\ 0.057092)$. Since Section 2.2.3 has passed the consistency check, it can be considered that the final weights obtained are reasonable. The convergence process is shown in Fig 5.

It can be seen from Fig 5 that the weight values of 6 evaluation indexes all fluctuate greatly in 50 times of simulation. However, after 150 times of stochastic simulation, the weight values of 6 evaluation indexes have no obvious fluctuation, which can be basically regarded as convergence. Therefore, as shown in Eq (5), this paper selects the average of 190 to 200 times of stochastic simulation as the final weight value.

**3.2.5 Debris flow risk prediction based on the cloud model.** The original cloud model is coupled by computing the membership of a single evaluation index [36,44]. Since the mean value between the two levels of the evaluation index is too sensitive, there will be situations that are difficult to define. Therefore, this paper proposes an improved hierarchical comprehensive cloud model for this problem. The specific implementation steps are as follows.

## (1) Normalize index parameters

Normalize the evaluation index data to eliminate the influence of dimension, as shown below:

$$b_{ij} = \begin{cases} \dfrac{r_{ij} - min_j(r_{ij})}{max(r_{ij})_j - min(r_{ij})_j} & (\text{Positive direction indicator}) \\[3mm] \dfrac{max_j(r_{ij}) - r_{ij}}{max_j(r_{ij}) - min_j(r_{ij})} & (\text{Inverse indicator}) \end{cases} \tag{6}$$

In the formula, $r_{ij}$ is the original evaluation index value of the sample, $b_{ij}$ is the normalized value.

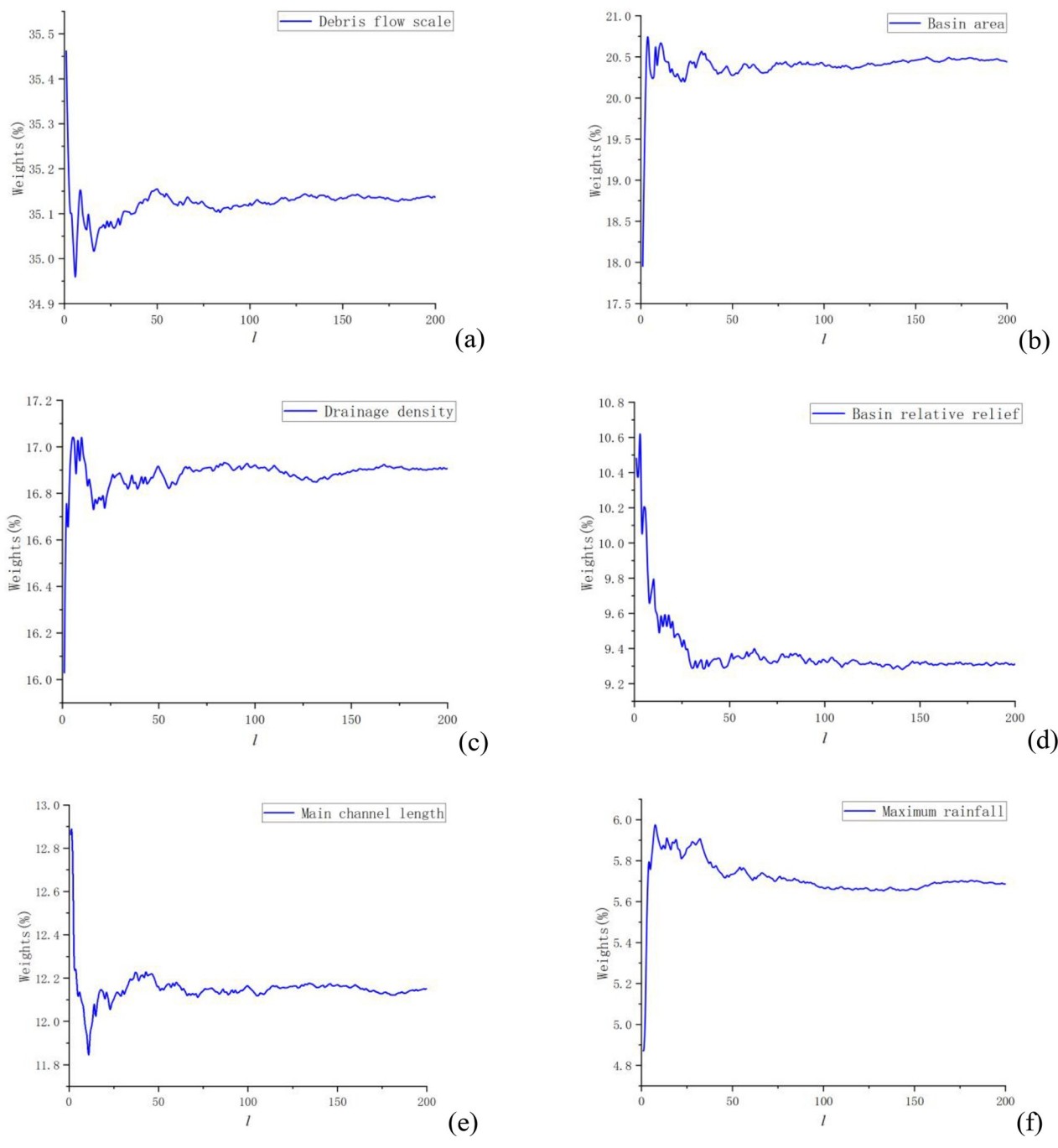

**Fig 5. The convergence process of each index.**

### (2) Generate a hierarchical comprehensive cloud

Enter the level boundary value for each evaluation index in Table 1, the cloud digital features of each evaluation index at each level are calculated. Due to the limited adaptability of the original cloud model calculation formula, and in order to ensure that the final comprehensive weight is fully covered at the grade boundary, combined with the classification characteristics of the single evaluation index of debris flow in Table 1, the following formula based on the

original cloud model is proposed:

$$\begin{cases} E_{x\mathrm{I}} = 0.5(\mathrm{max}b_{ij_\mathrm{I}} + \mathrm{min}b_{ij_\mathrm{I}}) \\ E_{n\mathrm{I}} = (\mathrm{max}b_{ij_\mathrm{I}} - \mathrm{min}b_{ij_\mathrm{I}})/3 \\ H_{e\mathrm{I}} = 0.001 \end{cases} \tag{7}$$

$$\begin{cases} E_{x\mathrm{II}} = 0.5(\mathrm{max}b_{ij_\mathrm{II}} + \mathrm{min}b_{ij_\mathrm{II}}) \\ E_{n\mathrm{II}} = (\mathrm{max}b_{ij_\mathrm{II}} - \mathrm{min}b_{ij_\mathrm{II}})/3 \\ H_{e\mathrm{II}} = 0.001 \times \dfrac{\mathrm{max}b_{ij_\mathrm{II}} - \mathrm{min}b_{ij_\mathrm{II}}}{\mathrm{max}b_{ij_\mathrm{I}} - \mathrm{min}b_{ij_\mathrm{I}}} \end{cases} \tag{8}$$

$$\begin{cases} E_{x\mathrm{III}} = 0.5(\mathrm{max}b_{ij_\mathrm{III}} + \mathrm{min}b_{ij_\mathrm{III}}) \\ E_{n\mathrm{III}} = (\mathrm{max}b_{ij_\mathrm{III}} - \mathrm{min}b_{ij_\mathrm{III}})/5 \\ H_{e\mathrm{III}} = 0.001 \times \dfrac{\mathrm{max}b_{ij_\mathrm{III}} - \mathrm{min}b_{ij_\mathrm{III}}}{\mathrm{max}b_{ij_\mathrm{I}} - \mathrm{min}b_{ij_\mathrm{I}}} \end{cases} \tag{9}$$

$$\begin{cases} E_{x\mathrm{IV}} = 0.5(\mathrm{max}b_{ij_\mathrm{IV}} + \mathrm{min}b_{ij_\mathrm{IV}}) \\ E_{n\mathrm{IV}} = (\mathrm{max}b_{ij_\mathrm{IV}} - \mathrm{min}b_{ij_\mathrm{IV}})/2.5 \\ H_{e\mathrm{IV}} = 0.001 \times \dfrac{\mathrm{max}b_{ij_\mathrm{IV}} - \mathrm{min}b_{ij_\mathrm{IV}}}{\mathrm{max}b_{ij_\mathrm{I}} - \mathrm{min}b_{ij_\mathrm{I}}} \end{cases} \tag{10}$$

In the formula, $\mathrm{max}b$ and $\mathrm{min}b$ represent the maximum and the minimum threshold value of the corresponding grade; such as $[\mathrm{min}b, +\infty]$, $[-\infty, \mathrm{max}b]$, the missing boundary can be determined according to the division rules of the actual data.

Then, the graded comprehensive cloud is introduced. It is obtained by synthesizing all the evaluation indexes belonging to the same grade of clouds. The synthesized formula is as follows [45]:

$$\begin{aligned} E_x(j) &= \sum_{i=1}^{n} E_x(i,j)E_n(i,j)w(i) / \sum_{i=1}^{n} E_n(i,j)w(i) \\ E_n(j) &= \sum_{i=1}^{n} E_n(i,j)w(i) \\ H_e(j) &= \sum_{i=1}^{n} H_e(i,j)E_n(i,j)w(i) / \sum_{i=1}^{n} E_n(i,j)w(i) \end{aligned} \tag{11}$$

In the formula, $w(i)$ ($i = 1,2,\cdots,n$) represents the weight of the $i$-th evaluation index, $n = 6$ in this paper; $j$ represents the number of grades, in this paper $j = I, II, III, IV$. $E_x(i,j)$ represents the expectation of the $i$-th evaluation index belongs to the $j$-th level cloud. $E_n(i,j)$ represents the entropy of the $i$-th evaluation index belonging to the $j$-th level cloud. $H_e(i,j)$ indicates the hyper-entropy of the $i$-th evaluation index belongs to the $j$-th level cloud. According to the classification boundaries given in Table 2 in Section 2.1, and the above calculation process, the final grade cloud for the prediction of debris flow risk is shown in Fig 6.

As can be seen from Fig 6, the comprehensive cloud model has complete coverage at all levels, which reduces the sensitivity of grading at the level boundary.

## (3) Determine the hazard level of debris flow

Referring to the research results of Shen et al., the cloud model can be considered as a popularization of precise numerical values. When the entropy $E_n$ and Hyper-entropy $H_e$ in the cloud model are both set to 0, the cloud model becomes a numerical value [25]. Therefore, in order to meet the same scoring scale as the previous comprehensive cloud, the cloud of all factors is synthesized to obtain the comprehensive expectation of the evaluation object. Calculation as follows:

$$Ex_b = \sum_{i=1}^{n} w(i) E_x(i) \tag{12}$$

In the formula, $w(i)$ represents the weight of the $i$-th evaluation index; $E_x(i)$ represents the expectation of the $i$-th evaluation index, which is the specific value $b$ of the evaluation index.

At this time, the grade comprehensive cloud calculated by Eq (11) and the comprehensive expectation calculated by Eq (12) are transformed on the same scoring scale. Inputting the forward cloud generator in Section 2.2.2, the membership degree of each level of the comprehensive expectation on the level comprehensive cloud can be obtained, and the hazard level of the debris flow sample to be predicted can be judged according to the principle of maximum membership degree.

## 4 Results

By inputting the data of Table 1 debris flow gully into the method proposed in section 3, the membership degree of each grade is obtained as shown in Table 4.

It can be seen from Table 4 that the improved cloud model has a more tendentious risk orientation, and the grade boundaries overlap proportionally according to the size of each grade range (as shown in Fig 5 above), so that the cases near the two grade boundaries also have a more accurate membership tendency. Therefore, the method in this paper can make a more accurate risk classification judgment for debris flow disasters, and can provide reference for the classification prevention of debris flow disasters.

In order to verify the reliability of the method, the calculation results of this paper are compared with the results of other literatures using different methods for the same case, as shown in Table 5.

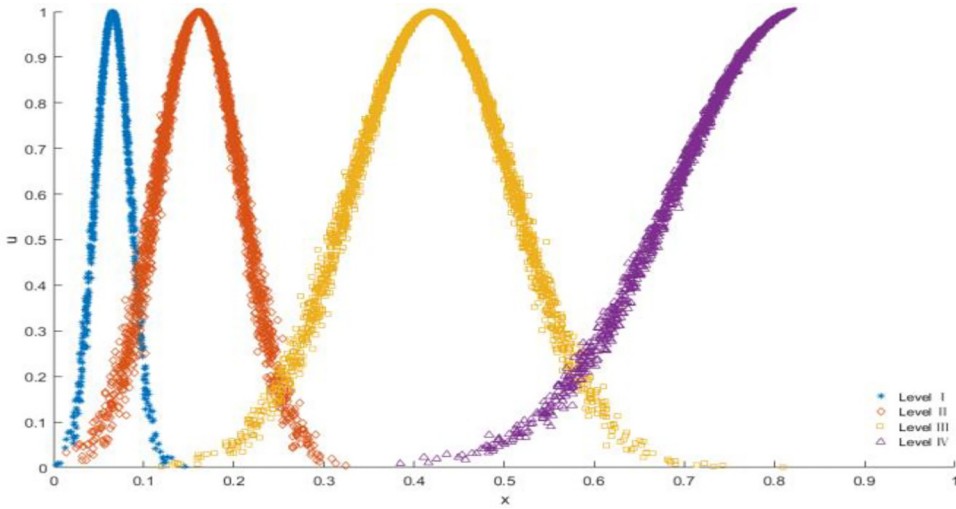

**Fig 6. Grade comprehensive cloud.**

**Table 4. Membership of each level.**

| Samples | Level I | Level II | Level III | Level IV | Results |
|---|---|---|---|---|---|
| A | 0 | 0 | 0 | 0.9999 | IV |
| B | 0 | 0 | 0 | 0.9999 | IV |
| C | 0 | 0.0200 | 0.3804 | 0 | III |
| D | 0 | 0 | 0.8983 | 0 | III |
| E | 0 | 0.6644 | 0.0730 | 0 | II |
| F | 0 | 0 | 0.1450 | 0.2927 | IV |
| G | 0 | 0 | 0.9936 | 0.0177 | III |
| H | 0 | 0.0984 | 0.2376 | 0 | III |
| Zhugongdi | 0 | 0.4613 | 0.1041 | 0 | II |
| Yindi gully | 0 | 0 | 0.4425 | 0.1247 | III |
| Jianshanbao | 0 | 0.0953 | 0.2405 | 0 | III |
| Xiushui creek | 0 | 0.2579 | 0.1535 | 0 | II |
| Shuilvqing | 0 | 0 | 0.9638 | 0.0240 | III |
| Shaoshui gully | 0 | 0.3142 | 0.1366 | 0 | II |
| Kuashanqing | 0 | 0.2724 | 0.1488 | 0 | II |
| Yanshui creek | 0 | 0.1696 | 0.1896 | 0 | III |
| Longtang gully | 0 | 0.3689 | 0.1230 | 0 | II |
| Lufang plain | 0 | 0.2260 | 0.1648 | 0 | II |
| Hujia gully | 0 | 0.0795 | 0.2565 | 0 | III |
| Laohu plain | 0 | 0 | 0.9385 | 0.0067 | III |

It can be seen from the analysis Table 5 that 2 of the 8 test debris flow gullies had evaluation results for the risk level that were one level higher than the results of other methods, and that 4 of the 12 example debris flow gullies had evaluation results that were one level higher than

**Table 5. Prediction results of different methods.**

| Samples | Ref. (Kuang at el. 2006 [6]) | Ref. (Li at el. 2017 [10]) | This paper's method |
|---|---|---|---|
| A | IV | IV | IV |
| B | IV | IV | IV |
| C | II | II | III |
| D | III | III | III |
| E | II | II | II |
| F | IV | IV | IV |
| G | III | III | III |
| H | II | II | III |
| Zhugongdi | | II | II |
| Yindi gully | | II | III |
| Jianshanbao | | II | III |
| Xiushui creek | | II | II |
| Shuilvqing | | III | III |
| Shaoshui gully | | II | II |
| Kuashanqing | | II | II |
| Yanshui creek | | II | III |
| Longtang gully | | II | II |
| Lufang plain | | II | II |
| Hujia gully | | III | III |
| Laohu plain | | II | III |

those proposed by Qiang et al [21]. This shows that the evaluation results of the method suggested in this study are more radical than those of the existing literature, which is obviously more conducive to the prevention and control of debris flow disasters. In addition, the evaluation results of the remaining samples are completely consistent with those of other methods, demonstrating the reliability of this method. The characteristics of this method will be described in the following discussion section.

## 5 Discussion

It can be seen from the above that the evaluation results of the proposed method are more radical than those of other methods. According to the principle of this method, the reasons are analyzed: in order to fully combine the advantages of various weight calculation methods, the weight is determined within a minimal convergence interval, which has a certain amount of fuzziness. Compared with the traditional cloud model which takes the minimum value of $H_e$, in the improved cloud model, $H_e$ is proportional to $E_n$, which increases the degree of atomization of the hierarchical cloud model. This is due to the consideration of the uniformity of cloud models at different levels and the fuzziness of the grading criteria. In addition, the improved comprehensive cloud model increases the overlapping parts at the levle boundaries and reduces the sensitivity between the level boundaries. In summary, the method proposed in this paper is more in line with engineering practice.

## 6 Conclusion

This paper presents a new method for debris flow risk assessment. The weight of the method is regarded as a random and evenly distributed variable within a reasonable interval, and its boundary is determined by fusing the weights of different methods with the characteristics of the cloud model. The rationality of the weight to be fused is tested in accordance with the characteristics of cloud model, and the final weight is calculated by the Monte Carlo Theory, which fully integrates the advantages of various calculation methods. The improved cloud model has more comprehensive coverage at the level boundary, reducing the difficult-to-define situation that occurs when the mean between two levels is oversensitive. Finally, the comparison between the evaluation results of the new method and those of other methods shows that the new method has higher reliability and can provide scientific guidance for the prevention and control of debris flow disasters.

In the following study, we will consider the combination of Geographic Information System (GIS) and the Remote Sensing (RS) technology, adding factors such as vegetation coverage or the area of pools zone, to better visualize the risk degree of debris flow disaster.

## Supporting information

**S1 Table. The data processing process of Fig 5.**
(XLSX)

**S1 Code. Cloud model and probability method code.**
(TXT)

## Author Contributions

**Conceptualization:** Shixin Zhang.

**Data curation:** Bo Ni.

**Formal analysis:** Bo Ni.

**Funding acquisition:** Li Li.

**Methodology:** Li Li.

**Resources:** Li Li, Yue Qiang.

**Software:** Shixin Zhang.

**Supervision:** Yue Qiang.

**Validation:** Dongsheng Zhao.

**Visualization:** Ling Zhou.

**Writing – original draft:** Bo Ni.

**Writing – review & editing:** Yue Qiang, Dongsheng Zhao, Ling Zhou.

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
