## [Decision Letter · Decision Letter 0]

5 Dec 2022

PONE-D-22-29284Risk Assessment of Debris Flow Disaster Based on Cloud Model - Probability Fusion MethodPLOS ONE

Dear Dr. li,

Thank you for submitting your manuscript to PLOS ONE. After careful consideration, we feel that it has merit but does not fully meet PLOS ONE’s publication criteria as it currently stands. Therefore, we invite you to submit a revised version of the manuscript that addresses the points raised during the review process.

We look forward to receiving your revised manuscript.

Kind regards,

Sher Muhammad, PhD

Academic Editor

PLOS ONE

Journal Requirements:

2. Please note that PLOS ONE has specific guidelines on code sharing for submissions in which author-generated code underpins the findings in the manuscript. In these cases, all author-generated code must be made available without restrictions upon publication of the work. 

Please review our guidelines at https://journals.plos.org/plosone/s/materials-and-software-sharing#loc-sharing-code and ensure that your code is shared in a way that follows best practice and facilitates reproducibility and reuse.

"This work was supported by the Scientific and Technological Research Program of Chongqing Municipal Education Commission (Grant No. KJQN202001218, KJQN202101206), the Open fund of Chongqing Three Gorges Reservoir Bank Slope and Engineering Structure Disaster Prevention and Control Engineering Technology Research Center (SXAPGC21ZD01), the Science and technology innovation project of Chongqing Wanzhou District Bureau of science and technology (wzstc20210305), Research development and application of “big data intelligent prediction and early warning cloud service platform for geological disasters in the Three Gorges Reservoir Area” of Chongqing Municipal Education Commission (HZ2021012), Special key program of Chongqing Technology Innovation and Application Development (Grant No. cstc2019jscx-tjsbX0015), and “Chongqing Huanjiang structure disaster prevention and reduction theory and key technology” of Chongqing University Innovation Research Group (201928), The guiding Science and Technology Plan Project of Chongqing Civil Architecture Society in 2022(2022A05, 2022B15, 2022B16)."

Please state what role the funders took in the study.  If the funders had no role, please state: ""The funders had no role in study design, data collection and analysis, decision to publish, or preparation of the manuscript."" If this statement is not correct you must amend it as needed. 

"This work was supported by the Scientific and Technological Research Program of Chongqing Municipal Education Commission (Grant No. KJQN202001218, KJQN202101206), the Open fund of Chongqing Three Gorges Reservoir Bank Slope and Engineering Structure Disaster Prevention and Control Engineering Technology Research Center (SXAPGC21ZD01), the Science and technology innovation project of Chongqing Wanzhou District Bureau of science and technology (wzstc20210305), Research development and application of “big data intelligent prediction and early warning cloud service platform for geological disasters in the Three Gorges Reservoir Area” of Chongqing Municipal Education Commission (HZ2021012), Special key program of Chongqing Technology Innovation and Application Development (Grant No. cstc2019jscx440 tjsbX0015), and “Chongqing Huanjiang structure disaster prevention and reduction theory and key technology” of Chongqing University Innovation Research Group (201928), The guiding Science and Technology Plan Project of Chongqing Civil Architecture Society in 2022(2022A05, 2022B15, 2022B16)."

"This work was supported by the Scientific and Technological Research Program of Chongqing Municipal Education Commission (Grant No. KJQN202001218, KJQN202101206), the Open fund of Chongqing Three Gorges Reservoir Bank Slope and Engineering Structure Disaster Prevention and Control Engineering Technology Research Center (SXAPGC21ZD01), the Science and technology innovation project of Chongqing Wanzhou District Bureau of science and technology (wzstc20210305), Research development and application of “big data intelligent prediction and early warning cloud service platform for geological disasters in the Three Gorges Reservoir Area” of Chongqing Municipal Education Commission (HZ2021012), Special key program of Chongqing Technology Innovation and Application Development (Grant No. cstc2019jscx-tjsbX0015), and “Chongqing Huanjiang structure disaster prevention and reduction theory and key technology” of Chongqing University Innovation Research Group (201928), The guiding Science and Technology Plan Project of Chongqing Civil Architecture Society in 2022(2022A05, 2022B15, 2022B16)."

7. PLOS requires an ORCID iD for the corresponding author in Editorial Manager on papers submitted after December 6th, 2016. Please ensure that you have an ORCID iD and that it is validated in Editorial Manager. To do this, go to ‘Update my Information’ (in the upper left-hand corner of the main menu), and click on the Fetch/Validate link next to the ORCID field. This will take you to the ORCID site and allow you to create a new iD or authenticate a pre-existing iD in Editorial Manager. Please see the following video for instructions on linking an ORCID iD to your Editorial Manager account: https://www.youtube.com/watch?v=_xcclfuvtxQ

8. We note that Figures 1 and 2 in your submission contain map images which may be copyrighted. All PLOS content is published under the Creative Commons Attribution License (CC BY 4.0), which means that the manuscript, images, and Supporting Information files will be freely available online, and any third party is permitted to access, download, copy, distribute, and use these materials in any way, even commercially, with proper attribution. For these reasons, we cannot publish previously copyrighted maps or satellite images created using proprietary data, such as Google software (Google Maps, Street View, and Earth). For more information, see our copyright guidelines: http://journals.plos.org/plosone/s/licenses-and-copyright.

(1) You may seek permission from the original copyright holder of Figures 1 and 2 to publish the content specifically under the CC BY 4.0 license. 

**Additional Editor Comments:**

The manuscript is generally well written and has the potential to publication. Before considering it for publication, the authors are advised to revise it as per the reviewers comments.

More important comments are:

Improve the language of the manuscript throughout. Improve the abstract to cover all components of the manuscript briefly and introduction sections.

Reviewers' comments:

Reviewer's Responses to Questions

**Comments to the Author**

1. Is the manuscript technically sound, and do the data support the conclusions?

Reviewer #1: Yes

Reviewer #2: Yes

2. Has the statistical analysis been performed appropriately and rigorously? 

Reviewer #1: Yes

Reviewer #2: Yes

3. Have the authors made all data underlying the findings in their manuscript fully available?

Reviewer #1: Yes

Reviewer #2: Yes

4. Is the manuscript presented in an intelligible fashion and written in standard English?

Reviewer #1: No

Reviewer #2: No

5. Review Comments to the Author

Reviewer #1: The manuscript addresses evaluating the risk of debris flow disasters based on the cloud model and the Monte Carlo method. The topic is important. However, there are some problems in the manuscript. Please take into consideration of the following comments to improve your manuscript for a major revision.

1. The language in the whole manuscript needs improvement. E.g., the last sentence in ‘Abstract’.

2. There are only 3 sentences in ‘Abstract’? Please improve it.

3. The work uses weight factor method. Entropy-based method is a common weight factor. I suggest the authors added some works of entropy-based weight factor for the convenience of the readership.

4. For example, in ‘Introduction’, please introduce some methods of entropy-based weight factor. E.g., ‘Measuring Uncertainty in the Negation Evidence for Multi-Source Information Fusion. Entropy. 2022; 24(11):1596. https://doi.org/10.3390/e24111596’.

5. Please highlight the motivation and contribution of using the cloud model and the Monte Carlo method simultaneously, e.g., in ‘Abstract’’ and ‘Introduction’.

6. I suggest the authors introduce the flow chart of the method (Figure 8) at the beginning of the subsection, not by the end of the section. This can help the readership find the main contribution of the work easily.

7. The details of some figures should be clearly described in the main text. E.g., please explain each sub-figure of ‘Fig. 6’ in the text.

8. The sequence number in Section 5 and 6 is not suggested. Complete paragraph is suggested.

Reviewer #2: This paper proposes a new method for risk assessment of debris flow gully based on Monte Carlo simulation and an improved cloud model. It is interesting. I suggest minor revision.

The English writing should be polished with help of professionals. There are some typos, grammatical errors and unsmooth expressions. For example, “literatures” should be corrected as “literature” in the 198th line on page 8.

On cloud model, some updated references should be commented to grasp the status of research. The following may be helpful: International Journal of Fuzzy Systems, 20(7) (2018) 2273-2300; Computer Modeling in Engineering & Sciences, 131(3) (2022) 1751-1792.

It would be better to add some solid comparative analyses in section 4 to explain the advantage of the proposed method of this paper.

Some future directions should be listed in section 6.

6. PLOS authors have the option to publish the peer review history of their article (what does this mean?). If published, this will include your full peer review and any attached files.

Reviewer #1: No

Reviewer #2: **Yes: **Shu-Ping Wan

---

## [Author Response · Author response to Decision Letter 0]

4 Jan 2023

Risk Assessment of Debris Flow Disaster Based on the Cloud Model - Probability Fusion Method

Li Li*, Bo Ni, Yue Qiang, Shixin Zhang, Dongsheng Zhao, Ling Zhou

Itemized responses to comments of academic editors and reviewers

The authors thank the editor and the reviewers’ support for publishing our manuscript. The authors are also grateful to the editor and the reviewers for the constructive suggestions that improved the manuscript. Based on the reviewers’ comments, the manuscript has been revised, and the revisions are marked in red. Our responses to the comments are presented below, numbered for ease of reference.

Journal Requirements:

1. Comment: Please ensure that your manuscript meets PLOS ONE's style requirements, including those for file naming. The PLOS ONE style templates can be found at 

Response: The author thanked the editor for the reminder, which has been revised in line with journal requirements.

2. Comment: Please note that PLOS ONE has specific guidelines on code sharing for submissions in which author-generated code underpins the findings in the manuscript. In these cases, all author-generated code must be made available without restrictions upon publication of the work. 

Please review our guidelines at https://journals.plos.org/plosone/s/materials-and-software-sharing#loc-sharing-code and ensure that your code is shared in a way that follows best practice and facilitates reproducibility and reuse.

Response: The author thanks the editor for the reminder that the code has been uploaded as a supplementary file.

3. Comment: We note that the grant information you provided in the ‘Funding Information’ and ‘Financial Disclosure’ sections do not match. 

Response: The author thanked the editor for the reminder, which has been revised in line with journal requirements.

4. Comment: Thank you for stating the following financial disclosure: 

~

Please state what role the funders took in the study. If the funders had no role, please state: ""The funders had no role in study design, data collection and analysis, decision to publish, or preparation of the manuscript."" If this statement is not correct you must amend it as needed. 

Response: The author thanks the editor for the reminder, which has been modified in cover letter.

5. Comment: Thank you for stating the following in the Acknowledgments Section of your manuscript: 

~

~

Response: The author thanks the editor for the reminder, which has been modified in cover letter.

6. Comment: In your Data Availability statement, you have not specified where the minimal data set underlying the results described in your manuscript can be found. PLOS defines a study's minimal data set as the underlying data used to reach the conclusions drawn in the manuscript and any additional data required to replicate the reported study findings in their entirety. All PLOS journals require that the minimal data set be made fully available. For more information about our data policy, please see http://journals.plos.org/plosone/s/data-availability.

Response: The author thanks the editor for the reminder, which has been modified in cover letter.

7. Comment: PLOS requires an ORCID iD for the corresponding author in Editorial Manager on papers submitted after December 6th, 2016. Please ensure that you have an ORCID iD and that it is validated in Editorial Manager. To do this, go to ‘Update my Information’ (in the upper left-hand corner of the main menu), and click on the Fetch/Validate link next to the ORCID field. This will take you to the ORCID site and allow you to create a new iD or authenticate a pre-existing iD in Editorial Manager. Please see the following video for instructions on linking an ORCID iD to your Editorial Manager account: https://www.youtube.com/watch?v=_xcclfuvtxQ

Response: The author thanks the editor for the reminder that the ORCID iD has been bound to the corresponding author.

8. Comment: We note that Figures 1 and 2 in your submission contain map images which may be copyrighted. All PLOS content is published under the Creative Commons Attribution License (CC BY 4.0), which means that the manuscript, images, and Supporting Information files will be freely available online, and any third party is permitted to access, download, copy, distribute, and use these materials in any way, even commercially, with proper attribution. For these reasons, we cannot publish previously copyrighted maps or satellite images created using proprietary data, such as Google software (Google Maps, Street View, and Earth). For more information, see our copyright guidelines: http://journals.plos.org/plosone/s/licenses-and-copyright.

Response: The author thanks the editor 's reminder, has deleted Fig. 1 and Fig. 2 and made a text description.

Reviewer #1:

The manuscript addresses evaluating the risk of debris flow disasters based on the cloud model and the Monte Carlo method. The topic is important. However, there are some problems in the manuscript. Please take into consideration of the following comments to improve your manuscript for a major revision.

1. Comment: The language in the whole manuscript needs improvement. E.g., the last sentence in ‘Abstract’.

Response: The reviewer’s comment is indeed and the authors thank the reviewer for this comment. The author has improved the grammar of the whole manuscript.

2. Comment: There are only 3 sentences in ‘Abstract’? Please improve it.

Response: The authors thank the reviewer for this comment. The author has improved the abstract section in lines 12 to 24.

3. Comment: The work uses weight factor method. Entropy-based method is a common weight factor. I suggest the authors added some works of entropy-based weight factor for the convenience of the readership.

Response: The authors thank the reviewer for this comment. The author has made relevant supplements in lines 51 to 58 of the introduction.

4. Comment: For example, in ‘Introduction’, please introduce some methods of entropy-based weight factor. E.g., ‘Measuring Uncertainty in the Negation Evidence for Multi-Source Information Fusion. Entropy. 2022; 24(11):1596. https://doi.org/10.3390/e24111596’.

Response: The reviewer’s comment is indeed and the authors thank the reviewer for this comment. The author has made relevant supplements in lines 51 to 58 of the introduction.

5. Comment: Please highlight the motivation and contribution of using the cloud model and the Monte Carlo method simultaneously, e.g., in ‘Abstract’’ and ‘Introduction’.

Response: The reviewer’s comment is indeed and the authors thank the reviewer for this comment. The author has re-discussed the whole abstract part and made relevant supplements in lines 59 to 73 of the introduction.

6. Comment: I suggest the authors introduce the flow chart of the method (Figure 8) at the beginning of the subsection, not by the end of the section. This can help the readership find the main contribution of the work easily.

Response: The authors thank the reviewer for this comment. The author has placed the flowchart at the beginning of Section 3.2 Model construction.

7. Comment: The details of some figures should be clearly described in the main text. E.g., please explain each sub-figure of ‘Fig. 6’ in the text.

Response: The authors thank the reviewer for this comment. The author has made the related supplementary explanation to Fig. 1 to Fig. 6.

8. Comment: The sequence number in Section 5 and 6 is not suggested. Complete paragraph is suggested.

Response: The authors thank the reviewer for this comment. The author has made corresponding modifications in lines 322 to 345.

Reviewer #2:

This paper proposes a new method for risk assessment of debris flow gully based on Monte Carlo simulation and an improved cloud model. It is interesting. I suggest minor revision.

1. Comment: The English writing should be polished with help of professionals. There are some typos, grammatical errors and unsmooth expressions. For example, “literatures” should be corrected as “literature” in the 198th line on page 8.

Response: The reviewer’s comment is indeed and the authors thank the reviewer for this comment. The author has improved the grammar of the whole manuscript.

2. Comment: On cloud model, some updated references should be commented to grasp the status of research. The following may be helpful: International Journal of Fuzzy Systems, 20(7) (2018) 2273-2300; Computer Modeling in Engineering & Sciences, 131(3) (2022) 1751-1792.

Response: The reviewer’s comment is indeed and the authors thank the reviewer for this comment. The author has made relevant supplements in lines 65 to 68 of the introduction.

3. Comment: It would be better to add some solid comparative analyses in section 4 to explain the advantage of the proposed method of this paper.

Response: The reviewer’s comment is indeed and the authors thank the reviewer for this comment. The author has made relevant supplements in section 4, lines 313 to 320.

4. Comment: Some future directions should be listed in section 6.

Response: The reviewer’s comment is indeed and the authors thank the reviewer for this comment. The author has added lines 343 to 345 in the conclusion.

---

## [Decision Letter · Decision Letter 1]

17 Jan 2023

Risk Assessment of Debris Flow Disaster Based on the Cloud Model - Probability Fusion Method

PONE-D-22-29284R1

Dear Dr. li,

We’re pleased to inform you that your manuscript has been judged scientifically suitable for publication and will be formally accepted for publication once it meets all outstanding technical requirements.

Kind regards,

Sher Muhammad, PhD

Academic Editor

PLOS ONE

Additional Editor Comments (optional):

I am pleased to recommend acceptance of the manuscript in the present form.

Reviewers' comments:

Reviewer's Responses to Questions

**Comments to the Author**

1. If the authors have adequately addressed your comments raised in a previous round of review and you feel that this manuscript is now acceptable for publication, you may indicate that here to bypass the “Comments to the Author” section, enter your conflict of interest statement in the “Confidential to Editor” section, and submit your "Accept" recommendation.

Reviewer #1: All comments have been addressed

Reviewer #2: All comments have been addressed

2. Is the manuscript technically sound, and do the data support the conclusions?

Reviewer #1: Yes

Reviewer #2: Yes

3. Has the statistical analysis been performed appropriately and rigorously? 

Reviewer #1: Yes

Reviewer #2: Yes

4. Have the authors made all data underlying the findings in their manuscript fully available?

Reviewer #1: Yes

Reviewer #2: Yes

5. Is the manuscript presented in an intelligible fashion and written in standard English?

Reviewer #1: Yes

Reviewer #2: Yes

6. Review Comments to the Author

Reviewer #1: Thanks for the revision and reply.

The paper has been improved.

I recommend acceptance of the manuscript.

Reviewer #2: The authors have revised the manuscript well according to my previous comments. It can be accepted now.

7. PLOS authors have the option to publish the peer review history of their article (what does this mean?). If published, this will include your full peer review and any attached files.

Reviewer #1: No

Reviewer #2: No

---

## [Editor Report · Acceptance letter]

19 Jan 2023

PONE-D-22-29284R1 

Risk Assessment of Debris Flow Disaster Based on the Cloud Model - Probability Fusion Method 

Dear Dr. li:

I'm pleased to inform you that your manuscript has been deemed suitable for publication in PLOS ONE. Congratulations! Your manuscript is now with our production department. 

Kind regards, 

on behalf of

Dr. Sher Muhammad 

Academic Editor

PLOS ONE